# Deep learning for twelve hour precipitation forecasts

Lasse Espeholt[1,2] ✉, Shreya Agrawal[1], Casper Sønderby[1], Manoj Kumar[1], Jonathan Heek[1], Carla Bromberg[1], Cenk Gazen[1], Rob Carver [1], Marcin Andrychowicz[1], Jason Hickey[1], Aaron Bell[1] & Nal Kalchbrenner [1,2] ✉

Existing weather forecasting models are based on physics and use super-computers to evolve the atmosphere into the future. Better physics-based forecasts require improved atmospheric models, which can be difficult to discover and develop, or increasing the resolution underlying the simulation, which can be computationally prohibitive. An emerging class of weather models based on neural networks overcome these limitations by learning the required transformations from data instead of relying on hand-coded physics and by running efficiently in parallel. Here we present a neural network capable of predicting precipitation at a high resolution up to 12 h ahead. The model predicts raw precipitation targets and outperforms for up to 12 h of lead time state-of-the-art physics-based models currently operating in the Continental United States. The results represent a substantial step towards validating the new class of neural weather models.

Probabilistic forecasts predict the likelihood of weather conditions at a given time and location. Weather conditions of interest can range from core atmospheric variables such as rate of rain and snow, wind velocity and direction, temperature, pressure levels, and solar coverage to weather patterns such as hurricanes, wildfires, and floods[1,2]. For the case of precipitation, a probabilistic forecast answers the question, "What is the current probability of a given amount of precipitation occurring at a location and time in the future?"

Short-term forecasting up to twelve hours in advance allows for predicting weather conditions with higher spatial and temporal precision than longer time ranges. This makes it possible for these forecasts to have substantial impact on society by helping with daily planning, energy management, transportation, and the mitigation of extreme weather events, among others[3]. Short-term forecasting is also a longstanding scientific challenge that combines our best understanding of the physics of the atmosphere with our most advanced computational capabilities. Current operational models for short-term forecasting are Numerical Weather Prediction (NWP) models that rely on physics-based simulations. The atmospheric simulations make use of supercomputers with heterogeneous hardware that run virtually continuously in data centers around the globe and update the

forecasts based on the latest observations. The weather conditions that the models predict include hundreds of atmospheric and oceanic features. The forecasts usually have a frequency of one or more hours and a grid resolution of 3–12 km. NWP methods obtain a probabilistic forecast by ensembling or post-processing the output of multiple individual physics-based models, each in turn requiring atmospheric simulation at the supercomputer scale. The accuracy of a physics-based forecast is tied to the grid resolution as more precise physics simulations require a finer representation of the state of the atmosphere. This relationship creates a computational bottleneck inherent to physics-based models that has proven challenging to overcome[4]. Besides resolution, the accuracy of the forecasts also depends on how well the physical models used in NWP describe the atmosphere at the various relevant scales; improving these models is a substantial scientific challenge by itself[5].

Due to the computational bottleneck, the large computational resources, and the time lag that physics-based models incur when making a forecast, efficient models based on deep neural networks represent a promising alternative framework for weather modeling[6,7] (Fig. 1). Instead of explicitly simulating the physics of the atmosphere, neural models learn the relationships between input observations and

[1]Google Research, Google Inc, 1600 Amphitheatre Pkwy, Mountain View, CA 94043, USA. [2]These authors contributed equally: Lasse Espeholt, Nal Kalchbrenner. ✉e-mail: lespeholt@google.com; nalk@google.com

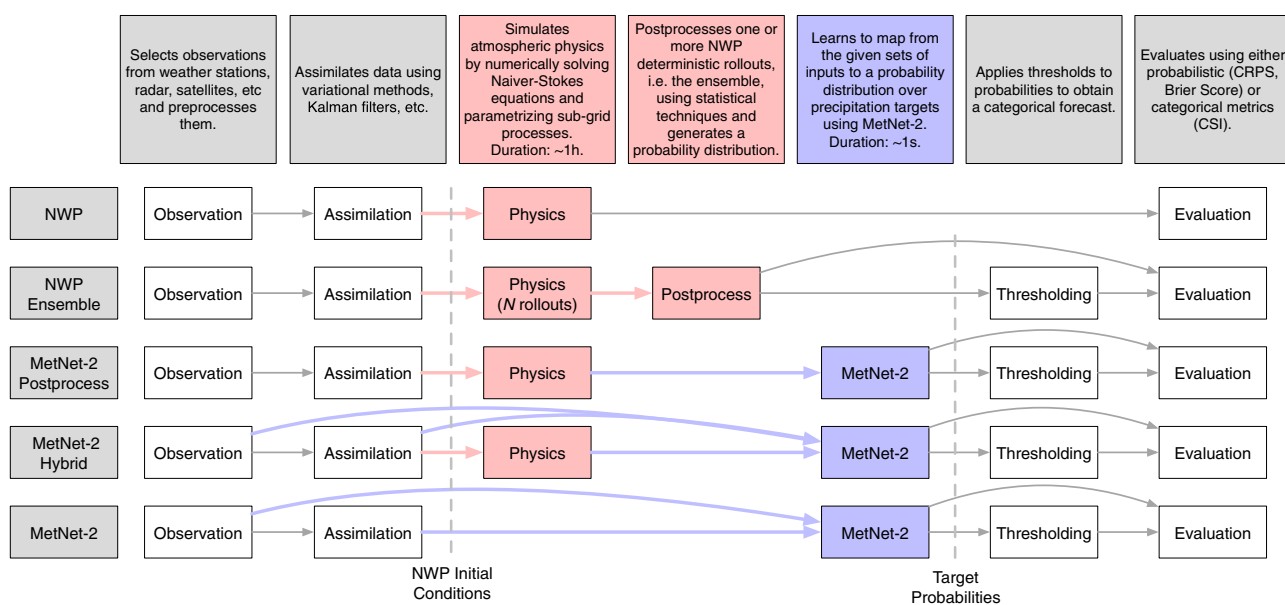

**Fig. 1 | Types of Weather Models.** Computation stages in a physics-based non-probabilistic NWP model (HRRR), a probabilistic ensemble NWP model (HREF), MetNet-2 and the two MetNet-2 variants. MetNet-2 does not rely on atmospheric simulation, whereas the other models do. The observation and assimilation phases prepare a representation of the current state of the atmosphere. Evaluation proceeds via categorical metrics such as CSI that require calculating decision thresholds or via probabilistic metrics such as CRPS.

output variables directly from data. Neural networks can run in a matter of seconds on parallel hardware and can thus generate forecasts more frequently and with higher spatial resolution. The networks are also notably simple and can be specified with generic modules in a few dozens of lines of code without hand-tuned routines for a specific task. The prediction of a neural network can also naturally be made probabilistic, learning to capture all the possible variability of the forecast from the data itself. These properties can not only offer improved forecasts, but also frequent and personalized forecasts[3] and open avenues of new applications that rely on the models' efficiency and flexibility. However, showing that the neural networks are able to learn to emulate the physics of the atmosphere sufficiently well to make skillful high-resolution forecasts for up to twelve hours ahead—a period that requires an advanced understanding of atmospheric physics and is well beyond the skill of extrapolation and short-term nowcasting methods[6,8–10]—is a substantial open challenge at the core of the neural modeling approach.

In this work, we present MetNet-2, a probabilistic weather model based on deep neural networks that is a successor to MetNet[7]. MetNet-2 features a forecasting range of up to 12 h of lead time at a frequency of 2 min and a spatial resolution of 1 km. In order to capture sufficient input context, MetNet-2 uses input observations from a 2048 km × 2048 km region and adopts novel neural network architectural elements in order to effectively process the large context. Such elements are (a) a context-aggregating module that enables the receptive field of the network to double after every layer, (b) a strong lead time conditioning scheme and (c) a model parallel training setup utilizing multiple chips for increased memory and parallel computation.

## Results

We train MetNet-2 to forecast precipitation, a fast-changing weather variable, over a 7000 km × 2500 km region of the Continental United States (CONUS). We find that MetNet-2 outperforms the probabilistic ensemble High-Resolution Ensemble Forecast (HREF) for the entire lead time range of 12 h according to the probabilistic metric Cumulative Ranked Probability Score (CRPS). When both MetNet-2 and HREF are thresholded to produce a categorical forecast, MetNet-2

outperforms HREF up to at least 9 h of lead time for both low and high rates of precipitation, according to the categorical Critical Success Index (CSI) metric. These results hold despite the key difference that MetNet-2 has an output resolution of 1 km and does not rely on forward atmospheric simulation, whereas HREF has a resolution of 3 km and relies on the results of five different forward atmospheric simulations from respective physics-based models, including those from the High-Resolution Rapid Refresh (HRRR) model[11].

We also study the performance of MetNet-2 in a hybrid mode where the physics-based forecast is used as an additional input to MetNet-2 itself. We find that Hybrid MetNet-2 is able to outperform a MetNet2-postprocessed HRRR forecast up to the entire range of 12 h, according to both CRPS and CSI. This shows the ability of MetNet-2 to extract and relay unique information that is not available in the atmospheric simulation even for longer lead times. MetNet-2's performance represents a step forward towards skillful forecasts with neural networks and suggests that MetNet-2 may be learning to emulate aspects of atmospheric physics. We perform an analysis into what MetNet-2 has learnt using state-of-the-art interpretation methods. The analysis reveals that MetNet-2 appears to make use of advanced physics principles when making its forecasts, which the model learns directly from the data.

**Comparison with HREF**

Although MetNet-2 uses the assimilation features that come from HRRR in order to gain a more complete picture of the initial state of the atmosphere, MetNet-2 does not rely on the atmospheric simulation itself that is the most computationally intensive part of an NWP model. The ensemble HREF model relies on 10 such simulations coming from five different NWP models each running on a supercomputer. The first result on dataset A is that MetNet-2 obtains a better CRPS than HREF over the entire lead time range of 12 h (Fig. 2). This metric is particularly appropriate in this evaluation as both CRPS and HREF are probabilistic. CRPS also takes into account the entire distribution across all precipitation rates; in other words, the result incorporates the performance from low to high rates of precipitation. When thresholding both MetNet-2 and HREF, optimized based on the categorical metric CSI, MetNet-2 outperforms HREF for at least the first 9 h of lead time

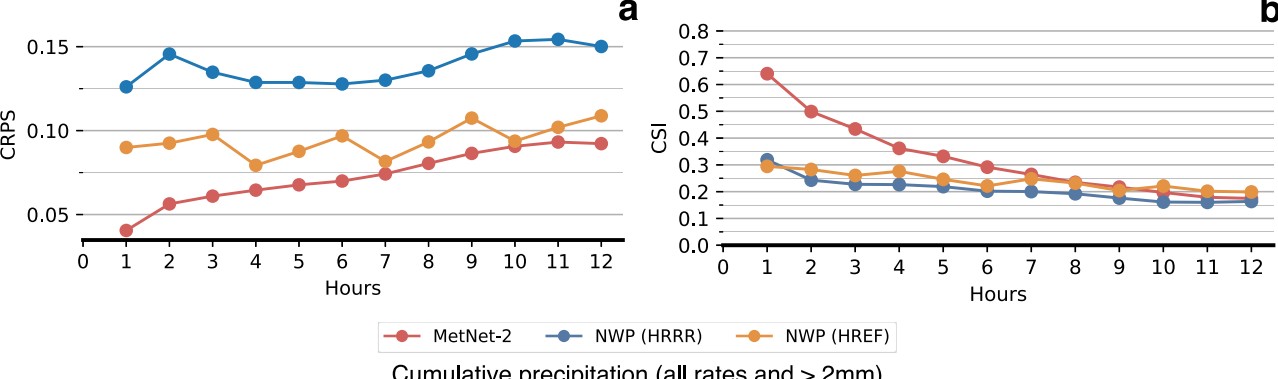

Cumulative precipitation (all rates and > 2mm)

**Fig. 2 | Evaluation of the models' performance (dataset A).** Performance comparison on test dataset A between the probabilistic MetNet-2 and HREF. Non-probabilistic HRRR results as reference. a Evaluation based on probabilistic CRPS that includes all rates of gauge-corrected hourly cumulative precipitation. **b** Evaluation based on the categorical CSI for the 2 mm/h rate (see Supplement E for other rates).

| Model / Hours | 1 | 2 | 3 | 4 | 5 | 6 | 7 | 8 | 9 | 10 | 11 | 12 |
|---|---|---|---|---|---|---|---|---|---|---|---|---|
| NWP (HRRR) | .32 | .24 | .23 | .23 | .22 | .20 | .20 | .19 | .18 | .16 | .16 | .16 |
| NWP (HREF) | .29 | .28 | .26 | .28 | .25 | .22 | .25 | .23 | .20 | **.22** | **.20** | **.20** |
| MetNet-2 | **.64** | **.50** | **.43** | **.36** | **.33** | **.29** | **.26** | **.23** | **.22** | .20 | .18 | .17 |
| NWP (HRRR) | .17 | .16 | .16 | .15 | .14 | .14 | .15 | .15 | .15 | .14 | .14 | .15 |
| MetNet-2 | **.44** | **.36** | **.32** | **.29** | .26 | .25 | .24 | .23 | .23 | .22 | .22 | .21 |
| MetNet-2 Postprocess | .30 | .26 | .25 | .24 | .23 | .23 | .24 | .24 | .24 | .24 | .24 | .24 |
| MetNet-2 Hybrid | **.44** | **.36** | **.32** | **.29** | **.28** | **.27** | **.26** | **.27** | **.26** | **.26** | **.25** | **.25** |

**Fig. 3 | Numerical results for the models (datasets A and B).** Critical Success Index scores of HRRR, HREF and MetNet-2 for cumulative precipitation on dataset A (first three rows) and of NWP (HRRR) and MetNet-2 variants for instantaneous precipitation of ≥2 mm/h on dataset B (last four rows). Scores are given for each of the 12 h of lead time. The best score for each set of results and each lead time is high-lighted in bold-face.

for low (0.2 mm/h) and high rates of precipitation up to 20 mm/h (Fig. 2 and Supplementary Fig. 3). MetNet-2 and HREF both outperform HRRR on these metrics across the whole 12 h range (Figs. 2 and 3). The skill gap between MetNet-2 and HREF is greatest in relative terms at the earliest hours and decreases gradually over time. Figure 4 represents two case studies of MetNet-2 and HREF forecasts. Both the uncertainty of the prediction and the variability grow over time and these aspects are evident in MetNet2's and HREF's forecasts. The probability of precipitation that MetNet-2 assigns to a given location tends to decrease on average over time as the probability mass is spread over a growing region of likely precipitation. The expected amount of precipitation that MetNet-2 forecasts in a patch closely matches the ground truth amount of precipitation.

### Hybrid results
A second core result is the performance comparison of MetNet-2 in a hybrid setting that uses the prediction of an NWP model, in this case HRRR. We compare MetNet-2 Postprocess with MetNet-2 Hybrid on test dataset B for both instantaneous and cumulative precipitation targets. MetNet-2 Postprocess maps HRRR's forecast to a probabilistic one. MetNet-2 Hybrid maps both MetNet-2's default inputs as well as HRRR's forecasts to a probabilistic forecast. Remarkably, the MetNet-2 architecture is able to add value to HRRR's postprocessed forecast all the way up to 12 h of lead time. That is, the performance of MetNet-2 Hybrid is higher than that of MetNet-2 Postprocess across the whole range based on both CRPS and CSI scores and for both low and high rates of precipitation (Figs. 3 and 5). Figure 6 shows a case study with these models and Fig. 7a, b visualize, respectively, the probabilistic error based on the Brier score achieved by the models, and the prediction regions for various rates of precipitation based on the CSI thresholds.

### Ablations
We perform various ablations on the default MetNet-2 in order to shed further light on the model's performance. A regional evaluation of MetNet-2 shows that the model performs well across diverse regions that see varying levels of annual precipitation (Supplement H). On the architectural front, we find that the size of the input context of 2048 km × 2048 km improves performance over context sizes of 1536 km × 1536 km, 1024 km × 1024 km and 512 km × 512 km (see "Methods" and Supplement F.1). The additional observations of the atmosphere that the assimilation process incorporates also have an impact on MetNet-2's performance, especially at later hours (see "Methods" and Supplement F.2). MetNet-2 is able to extract information from a broad range of observations and any additional observations are likely to improve MetNet-2's performance. Furthermore, both removing the special conditioning scheme for the lead time index (see "Methods" and Supplement F.4) and limiting the maximum dilation factor to 16 or below also impact MetNet-2's performance negatively (see "Methods" and Supplement F.5).

### Interpretation
MetNet-2's remarkable performance makes it important to understand what the physics-free neural network is learning. This can help researchers gain new insight about interactions between different meteorological variables and ensure that the model conforms to our prior knowledge about weather physics. We adopt a state-of-the-art neural interpretation technique called Integrated Gradients to attribute predictions to the input variables[12]. Among the notable findings, Fig. 8b shows that the relative importance of absolute vorticity is small for near-term forecasts, but grows in importance as lead time increases all the way up to 12 h. In Fig. 8a, the importance of upper-level vorticity for a twelve hour forecast is consistent with what is known as quasi-

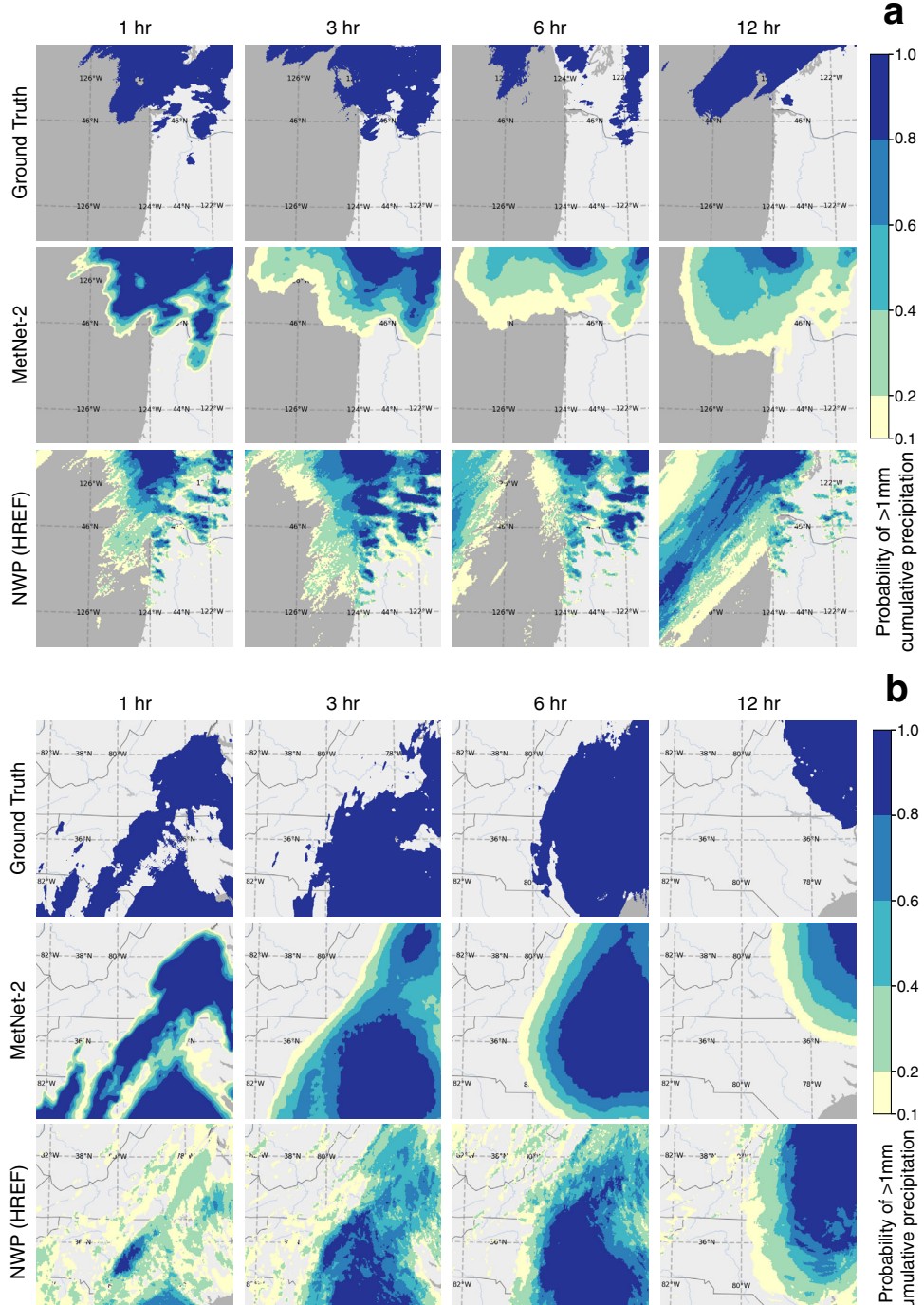

**Fig. 4 | Case studies for the models' forecasts of cumulative precipitation.** Forecasts for gauge-corrected hourly cumulative precipitation ≥1 mm/h. Each contour region corresponds to the band of probability of precipitation ≥1 mm/h that the respective model assigns to that region. **a** Case study for Thu Jan 03 2019 12:00 UTC of the North West coast of the US. **b** Case study of Hurricane Isaias, a Category 1 hurricane, that caused widespread destruction and economic damage. The forecast time is Mon Aug 03 2020 20:00 UTC on the East coast of the United States.

geostrophic theory, a non-trivial set of simplifications and filtering of the equations of motion. A key result in the theory is that positive vorticity in the upper-troposphere is consistent with upward motion in the lower-troposphere[13]. This upward motion does not directly trigger precipitation, but prepares the atmosphere for convection. See Supplement G for other key findings.

## Discussion

MetNet-2's strong performance for both low and high levels of precipitation and for both instantaneous and cumulative measures, its ability to estimate uncertainty and capture variation, its independence from atmospheric simulation, its design simplicity and the rapid and different nature of MetNet-2's computation represent a step towards a fundamental shift in forecasting from physics-based models to learning-based ones. The results also show how neural networks can learn to emulate complex and large-scale physics paving the way for ever more ambitious applications of neural nets in the physical sciences. Direct sensor data, although not readily available, can likely be used in place of the assimilation state to further reduce MetNet-2's total latency to essentially just the time required for observing the

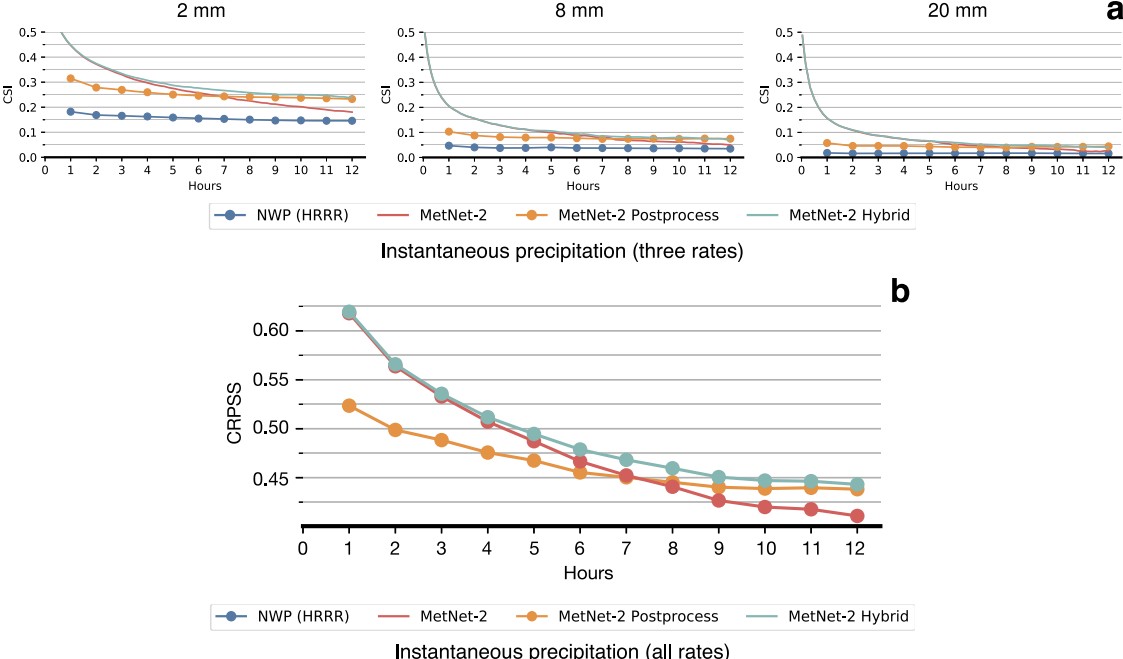

**Fig. 5 | Evaluation of the models' performance on low and high rates of precipitation (dataset B).** Tables for the second main comparison between the MetNet-2 variant Hybrid and Postprocessing using CSI and Cumulative Ranked Probability Skill Score (CRPSS). HRRR and MetNet-2 included for reference. **a** Critical Success Index metric for instantaneous precipitation of 2, 8, and 20 mm/h on test dataset B. **b** Continuous Ranked Probability Score Skill for instantaneous precipitation. The score tracks the relative improvement of MetNet-2, MetNet-2 Postprocess and MetNet-2 Hybrid over the HRRR reference.

atmosphere while removing any remaining reliance on NWP's initial state. Though designed for geo-spatial prediction, little in MetNet-2's architecture is specific to precipitation. This raises hopes that MetNet-2 could work well for many other weather variables possibly at once and even learn to transfer from one variable to the next and improve overall performance.

## Methods
### Framework
MetNet-2 and NWP models gather empirical observations in order to obtain an initial state of the atmosphere as a basis for their forecasts. Observations come from a variety of sensors that are located on the ground in weather stations, on satellites, on airplanes and balloons, and on ocean buoys, among others. An important source of observations in our framework are those coming from ground radars that densely populate the Continental United States. The reflectivity, measured by these radars, estimates the amount of precipitation at a given time and location. The estimates are made every few minutes and have a relatively high spatial resolution of 1 km × 1 km. In our framework, we use two types of precipitation measures: instantaneous precipitation that comes from the radar reflectivity at a temporal frequency of two minutes; and hourly cumulative precipitation that represents the amount of precipitation over the preceding hour. In the latter, rain gauges at weather stations are used to further corroborate the radar measurements improving the data reliability. The Multi-Radar Multi-System (MRMS) provides both of these measures[14]. While radar measurements provide information about the measures of precipitation, they do not describe the many other variables of the atmosphere, such as pressure, temperature and wind velocity and direction. Since the latter are not readily available, in order to incorporate them in our framework, we use the available set of atmospheric observations that result from the data assimilation process in the NWP model HRRR. This process uses various statistical and physics-based techniques to incorporate observations from the

atmospheric sensors including those coming from the radars themselves. The resulting state is the starting point for HRRR's simulation and we also adopt that state as an input for MetNet-2 to provide the model with more detailed information about the initial state of the atmosphere (for a full list of assimilation features, see ref. 15). In addition to radar and assimilation features, MetNet-2 also receives space-time coordinates for longitude, latitude, elevation, and forecast time[7] as well as optical satellite imagery; see Supplement B for a full description of data inputs.

### Ground truth
The radar precipitation measures are especially important for our task as they also serve as the ground truth training targets for MetNet-2. The instantaneous measures and the hourly cumulated measures are produced at 2 and 60 min intervals respectively. The measures range from a rate of 0–102.4 mm/h, with the higher and more extreme rates of precipitation becoming increasingly rare in the data; see Supplement B.2 summarizing how often various rates occur in the data.

### Dataset creation and splits
The data for MetNet-2 comes in input-output pairs where the inputs include radar, satellite, and weather state and outputs, the ground truth, correspond to the radar precipitation estimates. The available data spans a period from July 2017 to August 2020. The training, validation and test data sets are generated without overlap from periods in sequence. Successive periods of 400, 12, 40, 40 and 12 h are used to sample, respectively, training, validation, and test data, with the two 12 h periods inserted as hiatus. Spatially, the target patches are sampled randomly from intersections on a grid over the CONUS region spaced at .5 degrees in longitude and latitude. We sample two different test datasets, A and B, the former for our main comparison with HREF and the latter to compare the various MetNet-2 variants. Test dataset A covers only cumulative precipitation as HREF doesn't forecast instantaneous precipitation

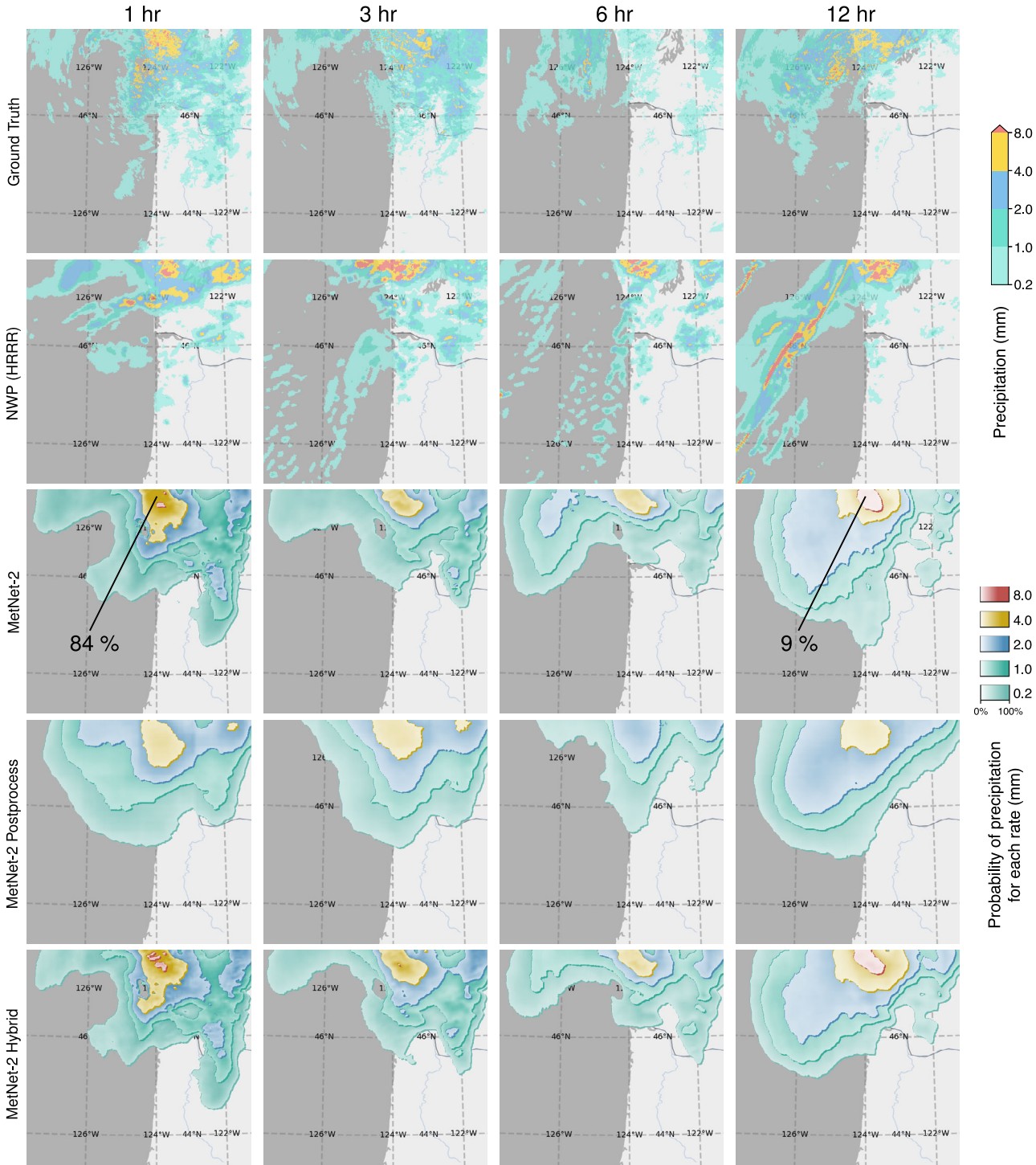

**Fig. 6 | Case study for the models' forecasts of instantaneous precipitation.**
Case study for Thu Jan 03 2019 12:00 UTC of the North West coast of the US with forecasts of instantaneous precipitation. Forecasts that use MetNet-2 are visualized as 3D-like contour plots. Each contour region corresponds to one of the visualised rates of precipitation of 0.2, 1.0, 2.0, 4.0, and 8.0 mm/h. The color intensity within the contour region that corresponds to rate $r$ is the probability of precipitation $P(\geq r)$ that the respective model predicts. Only probabilities above the corresponding CSI threshold are represented; these determine the contours. Contour regions are depicted overlapping one over the other starting from the lowest rate 0.2. We can observe in MetNet-2 models that the forecasted variability and the associated uncertainty grow with lead time. MetNet-2 Hybrid forecasts combine elements from both MetNet-2 and MetNet-2 Postprocess forecasts. The indicated percentage values denote forecast probabilities for the respective precipitation rate.

and the available HREF data over CONUS overlaps at 953 time-stamps with the rest of the data, from which the test dataset A is sampled. Test dataset B covers both cumulative and instantaneous precipitation and overlaps with the rest of the data at all time-stamps. Both datasets contain 39,841 patches each.

## MetNet-2 postprocess and hybrid
To study MetNet-2's performance in hybrid settings, we consider other training modes for MetNet-2 that, contrary to the default MetNet-2, make use of the outcome of NWP's atmospheric simulation. MetNet-2 Postprocess takes as an input HRRR's forecast for a given lead time

**Fig. 7 | Error analysis and additional rates in case study predictions.** Case study for Thu Jan 03 2019 12:00 UTC of the North West coast of the US with forecasts of instantaneous precipitation (same as Fig. 6). **a** Brier score maps that quantify the error of the probabilistic prediction. Lower error is better. The best scores are highlighted in bold-face. NWP's forecast is assigned probability 1 everywhere.

Scores are computed only over the region of high quality radar signal (Supplementary Fig. 1). **b** Decision boundaries based on thresholds optimally chosen for CSI. The numbers in each graphic correspond to the CSI score for the respective rate. Best scores are highlighted in bold face.

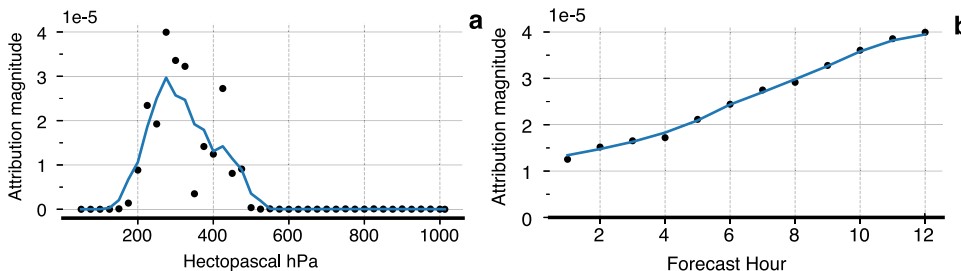

**Fig. 8 | Attribution analysis for MetNet-2.** Attribution of Absolute Vorticity, i.e., the amount a single input pixel contributes to a single pixel probability prediction. **a** Attribution of absolute vorticity at pressure levels 50hPa to 1013hPa on the 12 h

lead time prediction. The solid line is a 100hPa moving average over the attributions. **b** Attribution of absolute vorticity at 250 hPa on 1–12 h lead time predictions. The solid line is a 3 h moving average.

along with static location, altitude, and time features and learns to map HRRR's forecast as closely as possible to the ground truth. It also learns to correct for any systematic biases in HRRR's forecast and makes the forecast probabilistic. MetNet-2.

Hybrid learns to extract information from all the available inputs, including the twelve hourly forecasts from HRRR, as well as the radar and assimilation inputs used in the default MetNet-2. Whereas HRRR produces individual non-probabilistic rollouts, MetNet-2 and the variants are probabilistic at their core. Figure 1 summarizes the types of models and the respective steps.

## Model and architecture

A probabilistic forecast captures the combined uncertainty of both the measurements and the model:

$$P(\mathbf{r}_{x,y,t}|t_0) = f(\mathbf{c}_{x,y,t_0}, L) \tag{1}$$

where $\mathbf{r}$ are rates of precipitation, $x,y,t$ are the location and target time of the forecast, $t_0$ is the time at which the forecast is made, $\mathbf{c}_{x,y,t_0}$ is the atmospheric context at time $t_0$ relevant for location $x,y$ and $L = t - t_0$ is the lead time of the forecast. MetNet-2 bins the precipitation rates into

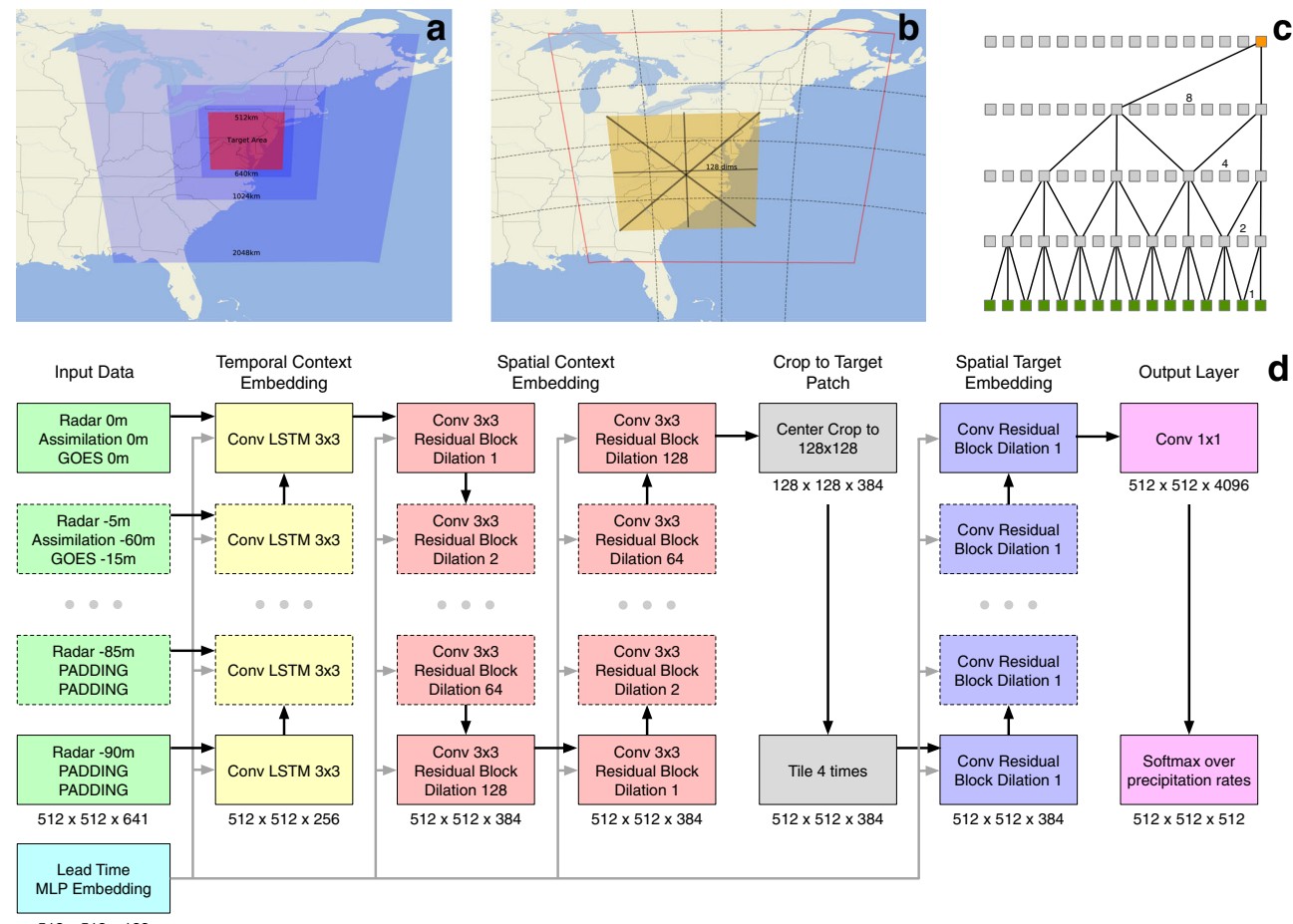

**Fig. 9 | MetNet-2 context aggregation and architecture.** Diagrams of various aspects of the MetNet-2 model. **a** MetNet-2 captures increasing amounts of context around the target patch. The figure shows the effects of the ortho-graphic projection for the context and target, onto Earth with an equir-ectangular projection. **b** MetNet-2 layers are spread over 4 × 4 TPU cores. The figure represents a convolution with dilation factor 128 reaching out to neigh-boring TPUs to retrieve the relevant parts of the layer. **c** Pattern of connectivity when using convolutions with dilation factors that double at each layer. The total receptive field grows exponentially in the number of layers. **d** MetNet-2 architecture. Radar, Assimilation, and Geo-spatial features enter the network along a time dimension. A convolutional recurrent network[10] embeds the input step by step. A stack of convolutional blocks with increasing dilation captures the large context of the embedded input. After a center crop corresponding to the target patch area and a tiling operation that restores the 1 km × 1 km reso-lution, a final stack of convolutional blocks produces a distribution over pre-cipitation levels for each target patch position. A rich embedding of the lead time index conditions each convolutional layer of the network.

512 categories that allow the model to forecast arbitrary discrete probability distributions over the categories[7].

The size of the input context plays a key role in the design of MetNet-2's architecture. Due to fast-changing nature of the atmo-sphere, the longer the lead time of the forecast for a location $x,y$ the more context the model needs around $x,y$ in order to have sufficient information for a skillful forecast. The context grows spatially in both dimensions and hence the total number of locations to attend grows quadratically in the length of the lead time. For a target patch of size 512 km × 512 km and forecast lead times of up to 12 h, MetNet-2 uses an input context size of 2048 km × 2048 km. This amounts to between 64 and 85 km of context per hour of lead time in each spatial dimension.

Besides making a large context available to the network, the network must be able to process and attend to the key parts of the context with its architecture. It is a special feature of the weather forecasting task that these key parts vary as a function of lead time: for the same input patch of data as lead time increases, the network must attend to key parts of an ever-growing potential region. These variable range dependencies present a challenge for the design of the neural architecture.

### Input encoder

The input to MetNet-2 captures 2048 km × 2048 km of weather con-text for each input feature, but it is downsampled via averaging by a factor of 4 in each spatial dimension, resulting in an input patch of 512 × 512 positions (Fig. 9a). The downsampling provides a trade-off between maintaining a sufficient amount of information in the context while substantially reducing the amount of computation required to encode this information.

In addition to the input patches having spatial dimensions, they also have a time dimension in the form of multiple time slices (see Supplement B.1 for details). This is to ensure that the network has access to the temporal dynamics in the input features. After padding and concatenation together along the depth axis, the input sets are embedded using a convolutional recurrent network[10] in the time dimension[7].

### Exponentially dilated convolutions

The next part of MetNet-2's architecture aims at connecting each position in the layer representing the encoded context with every other position in order to capture the full context. MetNet-2 uses two-dimensional convolutional residual blocks with a sequence of

exponentially increasing dilation factors of size 1, 2, 4,...., 128[16,17]. Dilation factors increase the receptive field of the convolution by skipping positions without increasing the number of parameters (Fig. 9c). Each position connects in this manner to all of the other 512 × 512 positions of the encoded tensor. Supplementary Fig. 2 illustrates the exact residual block with the dilated convolutions. Three stacks of 8 residual blocks form this context aggregating part of MetNet-2's architecture. The target patch of precipitation that MetNet-2 predicts corresponds to 512 km × 512 km and is centered in the middle of the 2048 km × 2048 km of the input patch. Because of that, the 512 × 512 positions from the context aggregation in the input encoder, are cropped to 128 × 128 positions. To obtain a prediction for the full size target patch, we upsample four times in each dimension, effectively creating another layer of 512 × 512 positions. This is processed with another shallow network and ends with a categorical prediction over 512 precipitation levels for each target position. See Fig. 9d for a full depiction of the architecture and Supplement D for additional architectural details.

### Conditioning with lead time

MetNet-2 encodes the lead time as a one-hot embedding with indices from 0 to 359 representing the range between 2 and 720 min[7] and mapped into a continuous representation. Instead of feeding the lead time embedding only at the input of MetNet-2, the embedding is applied both an additive and multiplicative factor to each of the two convolutional layers in the residual blocks of MetNet-2[18]. This ensures that the output of each convolutional layer now depends directly on lead time.

### Neural network parallelism

Due to the large input context, the 512 × 512 × $d$ input/internal representations and the 512 × 512 target patch, the network does not fit on a single TPU core. Instead of reducing the dimensions of the target patch, which will cause redundant computation since each smaller target patch will have overlapping input context, or reducing the dimensions of the internal representations, we use model parallelism. The input and the target is split into a four by four grid and processed by 16 interconnected TPU cores, with each TPU core responsible for a 128 × 128 area of the target, as shown in Fig. 9b. The necessary communication at each layer is handled automatically and efficiently[19,20]. This scheme that can be scaled further if needed makes it efficient to compute very large contexts for each target position.

## Data availability

The data sources used in this study—the MRMS[14], GOES[21] and HRRR[15] data—are publicly available for research use and can be accessed at the respective links.

## Code availability

We provide code for the MetNet-2 model and architecture that can be run with dummy inputs at https://colab.research.google.com/github/google/ai-weather-climate/blob/main/metnet2/colab.ipynb. See also pseudocode in Supplement D.

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

## Acknowledgements

We would like to thank Amy McGovern and Stephan Hoyer for insightful discussions and comments on the draft of the paper, and Zack Ontiveros, David McPeek, Ian Gonzalez, Claudio Martella, Samier Merchant, Fred Zyda, and Daniel Furrer for project and technical contributions.

## Author contributions

Research lead: N.K. Engineering lead: L.E. Data ingestion and dataset creation: L.E., S.A., C.G., and C.S. Model architecture: N.K., L.E., and M.K. Model training: N.K., L.E., J.H., M.K., C.S., and M.A. Model evaluation: L.E., N.K., and S.A. Model interpretation: S.A. Visualization: L.E., N.K., and S.A. Paper and revisions: N.K., L.E., and S.A. Technical advising and domain expertise: J.H., C.B., A.B., R.C., and M.A. Project and resource management: C.B., N.K., L.E., and A.B.

## Competing interests

The authors declare no competing interests.
