## [Peer Review File · Nature Communications]

Deep learning for twelve hour precipitation forecastsREVIEWER COMMENTS

Reviewer #1 (Remarks to the Author):

The authors present a study in which a neural network is used for short-term prediction of precipitation. The probabilistic model learns to predict precipitation patterns up to 12 hours ahead based on high-resolution observations. In another version of the model, it also uses high-resolution dynamical weather forecasts as input. The neural network predictions are compared to the aforementioned high-resolution model HRRR and favorable results are found at significant lead times.

This study presents a step forward in the application of neural networks for short-term precipitation forecasting. It builds upon previous work, most notably MetNet, by also incorporating other variables in the input, which, as the authors shows, has a significant positive impact on the prediction skill. Further, by merging the dynamical model predictions with the neural network, the authors combine the advantages of both approaches. The study also has a wide range of sensitivity experiments and generally shows a thorough evaluation.

However, from what I can tell, there is one major flaw in this study. The key result of the paper is an improvement in the CSI, BS and CRPS compared to HRRR. The proposed method is a probabilistic model whereas HRRR is a deterministic model. This is like comparing apples to oranges. On the CSI, a binary, point-wise metric, deterministic forecasts will suffer strongly from the double penalty problem for highly intermittent fields like precipitation. A thresholded probabilistic forecast will be much smoother and therefore would be expected to result in a better score. (This is why for deterministic forecasts, neighborhood scores like the FSS are now standard.) On the other hand, using probabilistic metrics like the BS and CRPS on a deterministic forecast is simply not fair and obviously the score of almost any decent probabilistic forecast will be significantly better. If HRRR was indeed the only baseline to be used, such evaluation would be ok, but this is not the case. First, there is a high-res ensemble system, HREF and, second, HRRR could be post-processed with rather simple techniques to improve its skill. I happened to be working on some HRRR/HREF evaluation at the moment, so in the paragraphs below I will demonstrate my arguments using data.

HREF is a multi-model ensemble that consists of 5 different models, each with another lagged version, so 10 members in total. Unfortunately, I was unable to find a preassembled version of HREF which is why I had to build the ensemble myself. Here I made one simplification to the operational version. For HRRR, I used a 12h lag (as for all the other models), instead of the operational 6h lag. The actual HREF, therefore, is likely a little bit better. Also, I am working with 6h accumulations, so much coarser in time compared to the present paper but I believe that my key results would be similar for different accumulation windows.

I computed the CSI for HRRR and HREF for a 20 mm / 6 h threshold for a 6-12 h lead time accumulation window (results are qualitatively similar for other thresholds). For HREF I used a probability threshold of 0.1 to convert the ensemble probabilities into a binary field (in fact, how to do this is another key question, see comment below). For HRRR I get a CSI of 0.075, for HREF 0.11, so around 50% better. Looking at the improvements in the present paper, they are quite a bit larger than 50% for most lead times but the difference between HREF and HRRR is quite significant. To illustrate that HRRR could be post-processed very simply to improve its scores, I created a simple 10 member ensemble based on randomly shifting HRRR around by 20 km horizontally for each member. This makeshift HRRR ensemble also got a better CSI (0.087) than the original HRRR model. I did not spend any time tuning this. I suspect that you can do even better quite easily.

In summary, I hope I have convinced the authors that the present evaluation is not adequate. I believe that an evaluation against HREF is the way to go for this study. The lead metric then could be the BSS instead of the CSI. As mentioned before, I think the results will still be strong against HREF.

On a technical note, a description of HREF can be found here

(<https://www.essoar.org/pdfjs/10.1002/essoar.10501462.1>) , data for the individual models is stored here: <https://data.nssl.noaa.gov/thredds/catalog/catalog.html>

Another key comment is on the availability of data and code. For any computational paper, the ability to look at the code is crucial for other researchers to build upon the study. I would strongly urge the authors to publish their code. At present, it would be very hard to impossible to exactly reconstruct the network structure from the paper. If, for some legal reason, this might not be possible, pseudo-code of the model architecture would also be acceptable. As for data, I understand that this is likely a huge amount of data. However, if the means exists to provide this data in an easily accessible cloud format, this would be a huge help to the community because some of the data can be hard to come by and download.

Other comments:

Line 6: I am not a science historian but I think the scientific study of weather has only really begun in the 19th century. This is also a very broad opening statement that doesn't really provide much relevant information.

Figure 1: It would be easier if the text was a little larger.

Section 2: I think the train/valid split has to be mentioned in the main text as it is crucial for ML studies.

Figure 2d: In the sketch of the NN design, it would be super helpful if you added the tensor size for each layer.

Line 175 and 397: How exactly do you determine the threshold to convert probabilistic forecasts to binary fields. Do you choose the threshold which gives you the best CSI on a validation set? How do you choose this validation set exactly?

Figure 4d: I don't think I quite understand what is shown here. Maybe this is connected to my confusion about the probability thresholding.

Line 355: How exactly is the train valid split done. This is crucial and should be described in detail.

Reviewer #2 (Remarks to the Author):

The presented work shows an interesting application of a complex neural network for probabilistic precipitation forecasting. The proposed network is compared with the predictions of a physics-based numerical model, resulting more accurate even for high forecast times, up to 12h. The results are important from several points of view, for the accuracy of the prediction, for the extension of the evaluated domain, for its resolution and for the original solutions used in the architecture of the neural network employed. An additional reason of interest lies in the ability to also integrate NWP model predictions in a hybrid configuration.

The work certainly represents a relevant contribution to the field of study, with interesting new elements compared to the state of the art. The analysis is conducted in a rigorous manner and described in sufficient accuracy and detail.

However, there are some aspects that should be further investigated, also with reference to what has been proposed in the recent literature.

In particular, it would be important to evaluate the performance of the model in terms of power spectral density of the predicted field at different spatial scales and for different forecast times. In similar systems there is a loss of signal definition for finer scales as the forecast time increases, this unintended feature of the neural network based forecast derives from the network optimization procedure which, while minimizing the loss function, devises "effective" smoothing of the solution. An analysis of the power spectral density for different forecast times also compared with the real data and with the HRRR model would be an interesting element that would allow to evaluate the effectiveness of the proposed solution.

In addition, the dataset used is of great interest and it would be useful if the authors could make public the same data, perhaps even highlighting a subset of the same, with cases of particular interest, to allow a comparison of results with other approaches and facilitate the advancement of research in this field.

Reviewer #3 (Remarks to the Author):

In this work the authors use neural networks to make precipitation forecasts over the continental USA, targeting short-range forecasting (0-12 hours). They test several forecasting paradigms: forecasting, post-processing forecasts and hybrid modelling incorporating data from both conventional forecasts and observations. By further developing the MetNet architecture the authors are able to improve predictions. On the longest assessed timescales the hybrid modelling produces the most accurate forecasts, however for most lead times the forecasting task is able to replicate the skill of the hybrid approach without using the time-evolved HRRR data. While some of the improvement over HRRR will stem from building a probabilistic output in contrast to the deterministic predictions of HRRR the differences between the results of the MNN Postprocess and other two networks show that the MNN is learning more than a probabilistic adaptation of the HRRR forecast. This is a very interesting piece of work, which not only builds a successful forecasting system but shines insight into how alternate forecasting paradigms compare. I also appreciated the section on interpretability.

Below I have a few minor questions/comments where I feel the authors could be clearer in their explanations and choices. After answering these I would be very happy for this submission to be accepted.

L134: What is the motivation and methodology for downsampling from 2048 grid points to 512 grid points?

L221: While I broadly agree with the conclusions, I think it's worth stating in the conclusions that none of the MNNs are fully physics-free as physics based models are currently necessary for the data-assimilation step. Perhaps in the long-term physics models can again be replaced here, but in this work the results still leverage physics models in each of the frameworks.

L369: Please include the exact assimilated variables provided to the networks including what pressure levels. Perhaps this could be presented as a table.

L389: Please be more specific in how you resample the data. Do you aim for a specific ratio of images with/without rain? It would have been interesting to see tests in how important this resampling was for performance.

P16 Polyak Decay: please could the authors provide more detail about this parameter.

L434: For accumulated precipitation training, how is this balanced in training. Is the network shown 30 2-minute instances for every 1h accumulated image?

Interpretability: This section is very interesting, but I have a few questions.

1. For Figures 21, 22. For each field at each pressure level each input pixel will contribute to the prediction of a single truth pixel. How do you aggregate over the input pixels, are we seeing the largest value, ie most important pixel?
2. I understand that attributions can be compared between fields. Are you showing the most important fields? i.e. is the attribution of 0.005 for the MRMS 0h data in figure 21 the largest average importance that you observe across all inputs?
3. Do the pressure values of the dots in fig 22 (and similar) correspond to the model levels used in the HRRR data assimilation system or has there been an interpolation step?

Figure 23: I think it would be helpful to plot the attribution of U and V with the same y-scale to aid comparison.

L623: "flow of horizontal wind towards the North". I think this is an error and the authors mean East.

Figure 24: Would it be possible to add some null test where we observe the attribution given to noise or an uncorrelated variable?

Figure 25: Is this the attribution for the prediction of a single pixel of rain or for the 128x128 target? If so could this be indicated and explained.

Response to Reviewers

Manuscript NCOMMS-21-34503-T

We kindly thank the reviewers for their insightful remarks. We start by listing the sections in the paper that received the most substantial updates:

- Title and abstract
- Results paragraph of Introduction (Section 1)
- Dataset splits (Section 2.2)
- Results (Section 4)
- Supplemental Results (Supplement E)
- Figures 1,2d,3,5,6,26 and Algorithm 1
- The model name was changed from “MNN” to “MetNet-2” throughout to indicate continuity with MetNet.

In the following we address the reviewers' remarks point-by-point.

Reviewer #1 (Remarks to the Author):

The authors present a study in which a neural network is used for short-term prediction of precipitation. The probabilistic model learns to predict precipitation patterns up to 12 hours ahead based on high-resolution observations. In another version of the model, it also uses high-resolution dynamical weather forecasts as input. The neural network predictions are compared to the aforementioned high-resolution model HRRR and favorable results are found at significant lead times.

This study presents a step forward in the application of neural networks for short-term precipitation forecasting. It builds upon previous work, most notably MetNet, by also incorporating other variables in the input, which, as the authors shows, has a significant positive impact on the prediction skill. Further, by merging the dynamical model predictions with the neural network, the authors combine the advantages of both approaches. The study also has a wide range of sensitivity experiments and generally shows a thorough evaluation.

However, from what I can tell, there is one major flaw in this study. The key result of the paper is an improvement in the CSI, BS and CRPS compared to HRRR. The proposed method is a probabilistic model whereas HRRR is a deterministic model. This is like comparing apples to oranges. On the CSI, a binary, point-wise metric, deterministic forecasts will suffer strongly from the double penalty problem for highly intermittent fields like precipitation. A thresholded probabilistic forecast will be much smoother and therefore would be expected to result in a better score. (This is why for deterministic forecasts, neighborhood scores like the FSS are now standard.) On the other hand, using probabilistic metrics like the BS and CRPS on a deterministic forecast is simply not fair and obviously the score of almost any decent probabilistic forecast will be significantly better. If HRRR was indeed the only baseline to be used, such evaluation would be ok, but this is not the case. First,

there is a high-res ensemble system, HREF and, second, HRRR could be post-processed with rather simple techniques to improve its skill. I happened to be working on some HRRR/HREF evaluation at the moment, so in the paragraphs below I will demonstrate my arguments using data.

HREF is a multi-model ensemble that consists of 5 different models, each with another lagged version, so 10 members in total. Unfortunately, I was unable to find a preassembled version of HREF which is why I had to build the ensemble myself. Here I made one simplification to the operational version. For HRRR, I used a 12h lag (as for all the other models), instead of the operational 6h lag. The actual HREF, therefore, is likely a little bit better. Also, I am working with 6h accumulations, so much coarser in time compared to the present paper but I believe that my key results would be similar for different accumulation windows.

I computed the CSI for HRRR and HREF for a 20 mm / 6 h threshold for a 6-12 h lead time accumulation window (results are qualitatively similar for other thresholds). For HREF I used a probability threshold of 0.1 to convert the ensemble probabilities into a binary field (in fact, how to do this is another key question, see comment below). For HRRR I get a CSI of 0.075, for HREF 0.11, so around 50% better. Looking at the improvements in the present paper, they are quite a bit larger than 50% for most lead times but the difference between HREF and HRRR is quite significant. To illustrate that HRRR could be post-processed very simply to improve its scores, I created a simple 10 member ensemble based on randomly shifting HRRR around by 20 km horizontally for each member. This makeshift HRRR ensemble also got a better CSI (0.087) than the original HRRR model. I did not spend any time tuning this. I suspect that you can do even better quite easily.

In summary, I hope I have convinced the authors that the present evaluation is not adequate. I believe that an evaluation against HREF is the way to go for this study. The lead metric then could be the BSS instead of the CSI. As mentioned before, I think the results will still be strong against HREF.

On a technical note, a description of HREF can be found here (<https://www.essoar.org/pdfjs/10.1002/essoar.10501462.1>), data for the individual models is stored here: <https://data.nssl.noaa.gov/thredds/catalog/catalog.html>

We thank the reviewer for this important observation that we agree with. The original version didn't include a probabilistic baseline as we found it difficult to obtain data for it. We have now included this comparison and put emphasis in the manuscript on two major experiments:

- A. MetNet-2 vs HREF, evaluation using CRPS and CSI, on hourly cumulative precipitation;
- B. MetNet-2 Hybrid vs MetNet-2 Postprocess, evaluation using CRPS and CSI, on hourly cumulative and instantaneous precipitation.

We have included HRRR as an informative reference in these comparisons. We find indeed that HREF is better than HRRR. The rephrasing affects many parts of the paper, including abstract, introduction and results sections in the main text and the appendix.

Another key comment is on the availability of data and code. For any computational paper, the ability to look at the code is crucial for other researchers to build upon the study. I would strongly urge the authors to publish their code. At present, it would be very hard to impossible to exactly reconstruct the network structure from the paper. If, for some legal reason, this might not be possible, pseudo-code of the model architecture would also be acceptable.

We have currently added pseudocode in the appendix for the model. We will also make the model code available through a shared notebook in such a way that dummy inputs can be passed through it (see Code Availability section).

As for data, I understand that this is likely a huge amount of data. However, if the means exists to provide this data in an easily accessible cloud format, this would be a huge help to the community because some of the data can be hard to come by and download.

We agree that such a benchmark would be very helpful to the community. Unfortunately, the large size of the data, the relative complexity of the infrastructure used to access it and to generate training and evaluation sets and, just as importantly, many licensing restrictions for the reproduction of the different data sources do not allow us to release the benchmark data. However, all the data is publicly accessible and can be downloaded independently by other researchers. We point this out in the Data Availability section.

Other comments:

Line 6: I am not a science historian but I think the scientific study of weather has only really begun in the 19th century. This is also a very broad opening statement that doesn't really provide much relevant information.

Done. We removed the broad opening statement.

Figure 1: It would be easier if the text was a little larger.

Done.

Section 2: I think the train/valid split has to be mentioned in the main text as it is crucial for ML studies.

Done. We have added Section 2.2 in the main text devoted entirely to this.

Figure 2d: In the sketch of the NN design, it would be super helpful if you added the tensor size for each layer.

Done.

Line 175 and 397: How exactly do you determine the threshold to convert probabilistic forecasts to binary fields. Do you choose the threshold which gives you the best CSI on a validation set? How do you choose this validation set exactly?

That's correct. We choose thresholds that give the best CSI on the validation set for each precipitation rate and each lead time. The added Section 2.2 also describes the creation of the validation set (random patches, non-overlapping time periods). The probabilistic CRPS metric does not rely on thresholds.

Figure 4d: I don't think I quite understand what is shown here. Maybe this is connected to my confusion about the probability thresholding.

We added a detailed description of what the "3d-like contour plots" represent. See caption to the figure (now Figure 6).

Line 355: How exactly is the train valid split done. This is crucial and should be described in detail.

We have added Section 2.2 in the main text and we have also added more details here in the appendix.

Reviewer #2 (Remarks to the Author):

The presented work shows an interesting application of a complex neural network for probabilistic precipitation forecasting. The proposed network is compared with the predictions of a physics-based numerical model, resulting more accurate even for high forecast times, up to 12h. The results are important from several points of view, for the accuracy of the prediction, for the extension of the evaluated domain, for its resolution and for the original solutions used in the architecture of the neural network employed. An additional reason of interest lies in the ability to also integrate NWP model predictions in a hybrid configuration.

The work certainly represents a relevant contribution to the field of study, with interesting new elements compared to the state of the art. The analysis is conducted in a rigorous manner and described in sufficient accuracy and detail.

However, there are some aspects that should be further investigated, also with reference to what has been proposed in the recent literature.

In particular, it would be important to evaluate the performance of the model in terms of power spectral density of the predicted field at different spatial scales and for different forecast times.

In similar systems there is a loss of signal definition for finer scales as the forecast time increases, this unintended feature of the neural network based forecast derives from the network optimization procedure which, while minimizing the loss function, devises "effective"

smoothing of the solution. An analysis of the power spectral density for different forecast times also compared with the real data and with the HRRR model would be an interesting element that would allow to evaluate the effectiveness of the proposed solution.

Thank you for the remark. We would like to make a few points here:

- We evaluate the models as to their ability to predict the target variables. We quantify the probabilistic error of the models with respect to the target (using CRPS and BS) and the models' ability to capture categorical precision/recall via metrics such as CSI. To the best of our understanding, the power spectral density doesn't evaluate the ability of the models to predict the targets so we do not find it directly relevant to support our main results.
- Probabilistic models are designed to capture all the variability and the uncertainty of the prediction. As such the predicted contours, depending on what is visualized, will naturally tend to be smoother. We think of this as a feature, not as a bug. We can see this in the HREF model too, that ensembles 10 rollouts and produces smoother regions of prediction as a result.
- Physics-based rollouts can have emerging phenomena that are informative from a meteorological perspective. In an end-to-end approach these phenomena could be predicted directly by the model (as supposed to be implicitly visualized by the rollout) - a process that could require additional annotation, but is beyond the scope of the current manuscript.

We hope that these points help address the remark and explain why we don't find evaluation with the power spectral density metric relevant for the manuscript's results.

In addition, the dataset used is of great interest and it would be useful if the authors could make public the same data, perhaps even highlighting a subset of the same, with cases of particular interest, to allow a comparison of results with other approaches and facilitate the advancement of research in this field.

Thank you for this remark. Please see the response to the same request to reviewer #1 above.

Reviewer #3 (Remarks to the Author):

In this work the authors use neural networks to make precipitation forecasts over the continental USA, targeting short-range forecasting (0-12 hours). They test several forecasting paradigms: forecasting, post-processing forecasts and hybrid modelling incorporating data from both conventional forecasts and observations. By further developing the MetNet architecture the authors are able to improve predictions. On the longest assessed timescales the hybrid modelling produces the most accurate forecasts, however for most lead times the forecasting task is able to replicate the skill of the hybrid approach without using the time-evolved HRRR data. While some of the improvement over HRRR will stem from building a probabilistic output in contrast to the deterministic predictions of HRRR the differences between the results of the

MNN Postprocess and other two networks show that the MNN is learning more than a probabilistic adaptation of the HRRR forecast. This is a very interesting piece of work, which not only builds a successful forecasting system but shines insight into how alternate forecasting paradigms compare. I also appreciated the section on interpretability.

Below I have a few minor questions/comments where I feel the authors could be clearer in their explanations and choices. After answering these I would be very happy for this submission to be accepted.

L134: What is the motivation and methodology for downsampling from 2048 grid points to 512 grid points?

Downsampling is performed via simple averaging of each 4 by 4 patch of features. The motivation is to encode the context more efficiently while still maintaining a good amount of information in the context. We rephrased and clarified this in the paper (line 164).

L221: While I broadly agree with the conclusions, I think it's worth stating in the conclusions that none of the MNNs are fully physics-free as physics based models are currently necessary for the data-assimilation step. Perhaps in the long-term physics models can again be replaced here, but in this work the results still leverage physics models in each of the frameworks.

Done. We have specified this throughout the manuscript.

L369: Please include the exact assimilated variables provided to the networks including what pressure levels. Perhaps this could be presented as a table.

The network includes all available HRRR initial state variables. We provide a link to a website with a table (https://home.chpc.utah.edu/~u0553130/Brian_Blaylock/HRRR_archive/hrrr_prs_table_f00-f01.html). Reference added to the manuscript (line 110).

L389: Please be more specific in how you resample the data. Do you aim for a specific ratio of images with/without rain? It would have been interesting to see tests in how important this resampling was for performance.

The resampling doesn't affect performance much, but it increases the convergence speed during training making model iterations somewhat faster. We made this clearer in the paper.

P16 Polyak Decay: please could the authors provide more detail about this parameter.

It is a parameter used for smoothing the effects of the stochastic gradient descent updates during training. Polyak decay or averaging keeps an exponentially-decaying running average of the weights of the network. At validation and test time, the averaged weights are used instead of

the weights from the very last update of the training procedure. See Tensorflow documentation for reference: https://www.tensorflow.org/api_docs/python/tf/train/ExponentialMovingAverage .

L434: For accumulated precipitation training, how is this balanced in training. Is the network shown 30 2-minute instances for every 1h accumulated image?

No. The network is shown on average the same number of hourly cumulative and instantaneous target patches during training.

Interpretability: This section is very interesting, but I have a few questions.

1. For Figures 21, 22. For each field at each pressure level each input pixel will contribute to the prediction of a single truth pixel. How do you aggregate over the input pixels, are we seeing the largest value, ie most important pixel?

We aggregate by taking the sum of the absolute values over all input pixels and all prediction pixels for each field at each pressure level. We then take an average of that over the input and prediction pixels to obtain the plots for each field at each pressure level. Hence, what we see here are the important fields on average for each input and prediction pixel.

2. I understand that attributions can be compared between fields. Are you showing the most important fields? i.e. is the attribution of 0.005 for the MRMS 0h data in figure 21 the largest average importance that you observe across all inputs?

That's correct, the attributions can be compared between fields and therefore the plots themselves can be compared with each other. It's also right that MRMS 0hr input is indeed that most important input across all inputs, which is also intuitively correct. Since we are predicting MRMS future values, the latest MRMS value should indeed provide the model with most information to make a prediction.

3. Do the pressure values of the dots in fig 22 (and similar) correspond to the model levels used in the HRRR data assimilation system or has there been an interpolation step?

The pressure values at the dots exactly correspond to the levels in HRRR data assimilation. They are 25 hPa apart.

Figure 23: I think it would be helpful to plot the attribution of U and V with the same y-scale to aid comparison.

Done. See new Figure 27.

L623: "flow of horizontal wind towards the North". I think this is an error and the authors mean East.

Corrected, thanks.

Figure 24: Would it be possible to add some null test where we observe the attribution given to noise or an uncorrelated variable?

This would require substantial retraining of the networks. But as observed in most neural networks, the weights attached to input that is random noise will likely go to zero as training progresses. Thus any attribution to that input noise should also be insignificant.

Figure 25: Is this the attribution for the prediction of a single pixel of rain or for the 128x128 target? If so could this be indicated and explained.

It is the attribution for the full 128x128 target. We clarified this in the caption.

REVIEWERS' COMMENTS

Reviewer #1 (Remarks to the Author):

I would like to thank the authors for their thorough revisions to the paper. From my own experience, I know that compiling the HREF data is not trivial, so I really appreciate the effort. I believe the paper is significantly stronger for it. This paper is an important contribution to the field. I recommend to accept the paper with a few minor comments below. Also, sorry for taking a long time with my review and delaying the process.

Stephan Rasp

Fig 1. In this figure I believe that the order of physics and ensembling for an NWP ensemble would make sense the other way around. Typically the ensembling, i.e. starting at different initial conditions and with different models happens before the simulation. As it's currently displayed readers might get the wrong idea that the ensembling for NWP ensembles happens after the physical simulations are run.

Section 4.1 "obtains a higher CRPS than HREF". This should be lower I believe.

CSI plots. I think it would help with the interpretability of the CSI plots if they all started at 0 on the y-axis.

Reviewer #2 (Remarks to the Author):

The authors have thoroughly addressed my comments, the paper is worthy of publication in its current form

Reviewer #3 (Remarks to the Author):

I thank the authors for their changes and updated manuscript. I would be happy to accept the manuscript for publication. Below I outline a few minor points that I would appreciate if the authors examine. I would like to congratulate the authors on an excellent publication.

Could the authors add the HREF values to the in table in fig 4a.

Very minor point, but in my view figure 22 would be improved by including zero. In all of these ablation figures it would be useful to have the MetNet2 value in the caption for context, e.g. mention context size of Met-Net2 was 2048.

Section D. This pseudo-code and diagram are very help. Could the authors expand upon the dense projections in D1. If I have understood the input for each projection is a scalar representing the lead-time and the output is a scalar to be used in either an additive or multiplicative manner. Is there a hidden state in this mapping of scalar to scalar? If so, how large.

Section G. Could the authors please clarify what they mean by baseline input x' , perhaps through use of an example. Thanks.

Response to Reviewers

Manuscript NCOMMS-21-34503-T

We kindly thank the reviewers for their insightful final remarks. We address each in turn.

Reviewer #1 (Remarks to the Author):

I would like to thank the authors for their thorough revisions to the paper. From my own experience, I know that compiling the HREF data is not trivial, so I really appreciate the effort. I believe the paper is significantly stronger for it. This paper is an important contribution to the field. I recommend to accept the paper with a few minor comments below. Also, sorry for taking a long time with my review and delaying the process.

Stephan Rasp

Fig 1. In this figure I believe that the order of physics and ensembling for an NWP ensemble would make sense the other way around. Typically the ensembling, i.e. starting at different initial conditions and with different models happens before the simulation. As it's currently displayed readers might get the wrong idea that the ensembling for NWP ensembles happens after the physical simulations are run.

Done. Thanks for the comment. We rephrased this part of the figure and hopefully it is clearer now.

Section 4.1 "obtains a higher CRPS than HREF". This should be lower I believe.

Done.

CSI plots. I think it would help with the interpretability of the CSI plots if they all started at 0 on the y-axis.

Done.

Reviewer #2 (Remarks to the Author):

The authors have thoroughly addressed my comments, the paper is worthy of publication in its current form

Reviewer #3 (Remarks to the Author):

I thank the authors for their changes and updated manuscript. I would be happy to accept the manuscript for publication. Below I outline a few minor points that I would appreciate if the authors examine. I would like to congratulate the authors on an excellent publication.

Could the authors add the HREF values to the in table in fig 4a.

Done.

Very minor point, but in my view figure 22 would be improved by including zero. In all of these ablation figures it would be useful to have the MetNet2 value in the caption for context, e.g. mention context size of Met-Net2 was 2048.

Done adding baselines in all the ablation study captions. We don't add results for time t0 in these specific diagrams due to consistency with all other diagrams as it would be substantial recomputing and editing work. HRRR's t0 results are somewhat lower than 100 due to the modifying effects of assimilation.

Section D. This pseudo-code and diagram are very help. Could the authors expand upon the dense projections in D1. If I have understood the input for each projection is a scalar representing the lead-time and the output is a scalar to be used in either an additive or multiplicative manner. Is there a hidden state in this mapping of scalar to scalar? If so, how large.

It is a 1-D vector to vector mapping. The lead time is converted to a one-hot projection of size #Lead time. A shallow MLP maps this to two vectors (one vector additive + one vector multiplicative) with dimensions equal to the hidden size of the residual network. The shallow MLP consists of a single layer and 512 hidden dimensions.

Section G. Could the authors please clarify what they mean by baseline input x' , perhaps through use of an example. Thanks.

Done.